# The Multi-Sites Trial on the Effects of Therapeutic Gardening on Mental Health and Well-Being

**DOI:** 10.3390/ijerph19138046

**Published:** 2022-06-30

**Authors:** Yeji Yang, Eunbin Ro, Taek-Joo Lee, Byung-Chul An, Kwang-Pyo Hong, Ho-Jun Yun, Eun-Yeong Park, Hye-Ryeong Cho, Suk-Young Yun, Miok Park, Young-Jo Yun, Ai-Ran Lee, Jeong-Ill Jeon, Songhie Jung, Tai-Hyeon Ahn, Hye-Young Jin, Kyung Ju Lee, Kee-Hong Choi

**Affiliations:** 1School of Psychology, Korea University, Seoul 02841, Korea; yaeji0917@korea.ac.kr (Y.Y.); lilyeunbin@korea.ac.kr (E.R.); 2KU Mind Health Institute, Korea University, Seoul 02841, Korea; 3Hantaek Botanical Garden, Yongin 17183, Korea; hantaek@hantaek.co.kr; 4Division of Forest & Landscape Architecture, Wonkwang University, Iksan 54538, Korea; askpp1048@wku.ac.kr; 5Korea Institute of Garden Design, Seoul 07995, Korea; hkp@dongguk.ac.kr; 6Landscape Yeoleum, Seoul 04026, Korea; hjyun@yeoleum.co.kr; 7Department of Environmental Landscape Architecture, Joongbu University, Geumsan 32713, Korea; eypark@joongbu.ac.kr; 8Seoul Green Trust, Seoul 04766, Korea; hyecho8149@naver.com; 9Department of Landscape Architecture, Daegu Catholic University, Gyeongsan 38430, Korea; yune1004@cu.ac.kr; 10Department of Smart Green City Industry Convergence, Korea Nazarene University, Cheonan 31172, Korea; ecoflower@naver.com; 11Department of Ecological Landscape Architecture Design, Kangwon National University, Chuncheon 24341, Korea; yyj@kangwon.ac.kr; 12Department of Human Environment Design, Cheongju University, Cheongju 28503, Korea; arlee@cju.ac.kr; 13Shingu Botanic Garden, Shingu College, Seongnam 13443, Korea; carpinus@shingu.ac.kr; 14Gardens and Education Research Division, Korea National Arboretum, Pocheon 11186, Korea; jungsonghie@korea.kr (S.J.); ahnthez@korea.kr (T.-H.A.); 15Integrative Obstetrics & Gynecology, Institute for Occupational & Environmental Health, Korea University, Seoul 02841, Korea

**Keywords:** gardening, nature-based intervention, psychosocial intervention, mental health, well-being, public health, COVID-19

## Abstract

Although many people affected by COVID-19 suffer from some form of psychological distress, access to proper treatment or psychosocial interventions has been limited. This study aimed to examine the feasibility and preliminary effects of a therapeutic gardening program conducted during the COVID-19 pandemic. The program consisted of 30 sessions and was conducted at 10 nationwide sites in Korea from June to November 2021. Mental health and well-being were assessed using the Mental Health Screening Tool for Depressive Disorders, Mental Health Screening Tool for Anxiety Disorders, Engagement in Daily Activity Scale, brief version of World Health Organization Quality of Life, and Mindful Attention Awareness Scale. Cohen’s *d* value was calculated for the effect size, and a multilevel analysis was used to determine the longitudinal effects of therapeutic gardening. The effect sizes for depression, anxiety, daily activities, quality of life, and mindfulness were 0.84, 0.72, 0.61, 0.64, and 0.40, respectively. Multilevel analyses showed that all five mental health variables improved significantly over time as the therapeutic gardening program progressed. Therapeutic gardening is promising and applicable as a nature-based intervention to improve the mental health of individuals experiencing psychological distress especially in the COVID-19 pandemic.

## 1. Introduction

An increasing number of people in modern society experience psychological distress and mental disorders that cause significant socioeconomic burdens [1,2,3]. Despite the high prevalence of mental disorders such as depression (12.9%) and anxiety (7.3%), most people (63.2–86.3% depending on the countries) do not have proper access to optimal psychosocial treatment [4,5,6]. Moreover, as the COVID-19 pandemic has progressed, depression and anxiety have become increasingly serious psychological problems for people who were infected by the COVID-19 or the general public having pre-existing mental health issues [7,8,9,10]. More than half of the world population was affected by COVID-19 at a moderate-to-severe level of psychological impact, and 14–38.2% of the public reported depression or anxiety symptoms during the COVID-19 pandemic [11,12,13].

Especially, compared to pre-COVID-19, the public’s mental health such as depression and anxiety has significantly deteriorated [14,15,16]. Lockdown and social isolation policy led the public to stay indoors and deprived them of protective factors such as physical activities and social relationship [13,15]. Under the stressful situation, people with mental health issues or vulnerability to stress (e.g., neuroticism [17]) could be easily influenced by the pandemic, leading to relapses or worsening of existing mental illnesses [18]. Therefore, an accessible and feasible therapeutic intervention for people with psychological distress during the COVID-19 pandemic is required [8].

Many studies have shown that greater engagement with nature is linked to better mental health, such as low depression and stress levels and high sleep quality [19,20,21,22,23]. The effectiveness and applicability of nature-based therapy as an intervention for public health are supported by several systematic reviews with reliable evidence [24,25,26]. Therapeutic gardening is a form of nature-based activity and cultivation of plants that promote mental and physical health and well-being [25,26,27,28,29,30,31]. Previous studies have reported on the effect of gardening on alleviating symptoms of mental disorders (e.g., schizophrenia and attention deficit hyperactivity disorder) [32,33,34] and promoting positive mental health such as mindfulness, positive emotions, well-being, vigor, life satisfaction, and quality of life (QoL) [29,35,36,37,38,39,40,41]. In addition, some studies have shown that patients with physical disorders, such as obesity or cancer, and those with mental disorders benefit from gardening [38,42,43].

Although previous studies have reported the positive effects of gardening, the effects varied depending on the individual characteristics (e.g., age and presence of mental or physical disorders) of the participants [44,45,46]. For instance, patients with dementia, depression, or cardiac rehabilitation showed twice the effect size compared to studies with non-patients [45]. Men also showed greater changes in anxiety than women [47]. However, most previous studies had a small sample size and failed to include multiple subgroups. When participating in the identical gardening program, the difference in the effects among participants remains unclear. Several recent studies showed gardening was associated with lower psychological distress and higher mental resilience during COVID-19 [48,49]. However, no studies with a larger sample size have reported the effects of therapeutic gardening on mental health during the COVID-19 pandemic. 

In this study, we have two objectives. First, we aimed to investigate the therapeutic effect of a gardening program on the mental health and well-being of people with mild-to-moderate levels of depression and anxiety during the COVID-19 pandemic with a large sample size. We hypothesized that the mental health and well-being of participants would significantly improve after therapeutic gardening program. Second, we examined whether the benefits of the therapeutic gardening program differed according to participants’ characteristics such as gender, age, and the presence of mental disorder.

## 2. Materials and Methods

### 2.1. Participants 

Participants were screened using the Mental Health Screening Tool for Depression (MHS:D) [50] and Mental Health Screening Tool for Anxiety (MHS:A) [51]. Those with scores higher than the MHS:D score of 8 or MHS:A score of 10, indicating mild depression and anxiety levels, respectively, were included in the study. A total of 111 participants who experienced depression or anxiety symptoms were recruited from 10 centers nationwide, including local community centers, mental health centers, senior welfare centers, dementia daycare centers, hospitals, and special schools. To recruit participants, a flyer containing the research purpose, contents of the therapeutic gardening program, and measurements, was posted on bulletin and online. Participants’ diagnoses were obtained from psychiatrists or mental health professionals who attended the local mental health centers. The exclusion criteria included mobility problems, communication difficulties, or severe mental illnesses. All participants received both oral and written explanations of the study and provided written informed consent prior to the pre-assessment. For compensation, mental health report was provided for each participant. This study was approved by the local institutional review board.

### 2.2. Procedure

We employed a longitudinal and prospective design to examine the effects of a therapeutic gardening program on participants’ mental health and well-being. Participants’ depression, anxiety, and psychological vitality were measured once every 2 weeks for a total of eight sessions (sessions 1, 5, 9, 13, 17, 21, 25, and 30). Including depression, anxiety, and vitality, a total of five self-rated mental health and well-being measures were measured twice: at baseline (at the first session) and at the end of the program (at the last session). The therapeutic gardening program was administered as a group for 15 weeks (twice per week for a total of 30 sessions). Each session required an average of 3 h, including preparation, assessments, warm-up, main gardening activities, and group feedback. An example of a therapeutic gardening program is presented in Appendix A. The participants were divided into two or three groups to reduce the number of participants as a quarantine policy and for more flexible and individualized gardening activities. All gardening activities were conducted outdoors (except for bad weather) for therapeutic purposes, such as observing plants, creating a garden, and taking a walk in the garden. 

### 2.3. Measures

#### 2.3.1. Mental Health Screening Tool for Depressive Disorders 

The Mental Health Screening Tool for Depression (MHS:D) is a self-reported assessment tool for early screening and intervention in patients with major depressive disorder in a primary medical setting, with relatively high accuracy [50]. It consists of 12 items that ask how much an individual has experienced symptoms related to major depressive disorder in the past 2 weeks. MHS:D is rated on a 5-point Likert scale from 0 to 4, with a high sum score indicating more severe depression. A total score of 0–8 is considered the minimal range, 8–12 is mild, 12–20 is moderate, and over 20 is severe. The MHS:D was developed in two versions, online and offline, and both versions were used in this study, which was more convenient for the participants. In a validation study, the MHS:D showed excellent internal consistency [50].

#### 2.3.2. Mental Health Screening Tool for Anxiety Disorders 

The Mental Health Screening Tool for Anxiety (MHS:A) [51] is a self-reported assessment tool with relatively high accuracy for early screening and intervention in patients with generalized anxiety disorder in the primary medical setting. It consists of 11 questions that ask how often an individual has experienced symptoms related to generalized anxiety disorder in the past 2 weeks. MHS:A is rated on a 5-point Likert scale from 0 to 4, with a high sum score indicating more severe anxiety. A total score of 0–10 is considered the minimal range, 10–20 is mild, 20–30 is moderate, and over 30 is severe. MHS:A was developed in two versions, online and offline, and both versions were used in this study, which was more convenient for participants. In a validation study, the MHS:A showed excellent internal consistency [51].

#### 2.3.3. Engagement in Daily Activity Scale 

The Engagement in Daily Activity Scale (EDAS) is a self-reported assessment of the level of daily activity [7]. It consists of five items that investigate the extent to which an individual has engaged in daily activities over the past week. The EDAS is rated on a 5-point Likert scale from 1 to 5, with a high sum score indicating more activity in daily life. In a preliminary study on the psychological impact of COVID-19 in South Korea, the EDAS showed good internal consistency [7].

#### 2.3.4. Korean Version of World Health Organization Quality of Life–Brief 

The World Health Organization Quality of Life—Brief version (WHOQOL-BREF) is a self-report assessment tool used to measure the QoL [52]. The WHOQOL-BREF consists of 26 items rated on a 5-point Likert scale ranging from 1 to 5. The WHOQOL-BREF consists of two items asking about overall QoL and satisfaction with health, and comprises four sub-domains: physical health, psychological health, social relationships, and environment. A high score indicated high QoL. In the current study, the validated Korean version of the WHOQOL-BREF was used [53]. In a validation study of the Korean version of the WHOQOL-BREF, a satisfactory level of internal consistency and a significant level of test-retest reliability were reported.

#### 2.3.5. Korean Version of Mindful Attention Awareness Scale 

The Korean Version of Mindful Attention Awareness Scale (MAAS) is a self-reported assessment tool used to measure attention and awareness, which are the core constituent concepts of mindfulness [54]. The MAAS consists of 15 items rated on a 6-point Likert scale ranging from 1 to 6. A high score indicates greater mindfulness. In this study, a validated Korean version of the MAAS was used [55]. A very good level of internal consistency was reported in a validation study of the Korean version of MAAS.

### 2.4. Statistical Analysis

Statistical analyses were conducted by an analytical research team, independent of the therapeutic gardening teams. Cohen’s *d*, one of the most common ways to measure effect size, was calculated for each of the five mental health measures (MHS:D, MHS:A, EDAS, WHOQOL-BREF, and MAAS). Cohen’s *d* refers to Cohen’s *d_z_* in this study, which is the recommended effect size for one sample or correlated sample comparisons [56]. Cohen’s *d* can be interpreted as small (*d* = 0.2), medium (*d* = 0.5), or large (*d* = 0.8) effect sizes based on Cohen’s suggestion [57]. To examine the effects of therapeutic gardening on mental health and well-being, a multilevel analysis (a type of hierarchical linear mixed model) was conducted. A multilevel analysis has several advantages in longitudinal analyses; it does not require a balanced design or equally spaced measurement and can allow for clustering at a higher level, enabling simultaneous modeling of both intra- and inter-individual changes [58,59]. In this study, three-level modeling was adopted because the collected data were heterogeneous according to the site and number of assessments. Time was included as a within-subjects (Level 1) parameter, whereas individuals (Level 2) and groups (Level 3) were included as between-subjects parameters. Group-mean centering was used to reduce the risk of multicollinearity and increase the ease of interpretation [58]. 

A moderation analysis was conducted to investigate subgroup differences in the effect of therapeutic gardening on the five variables of mental health and well-being (depression, anxiety, daily activities, quality of life, and mindfulness). Gender, age, and presence of mental disorders were considered as categorical moderators. Age was classified as elderly population (older adults) and adults based on the age of 65 according to the organization for economic co-operation and development (OECD) age classification [60]. Statistical analysis was conducted using R software Version 4.1.2 (which was developed by RStudio Team in Boston, MA, USA) using the “lmerTest” [61], “lme4” [62], and “nlme” [63] packages. To facilitate the interpretation of the results, the “ggplot2” package [64] was used for visualization.

## 3. Results

The characteristics of the participants in relation to demographics, mental disorders, and education level are presented in Table 1. A total of 111 participants were recruited from local community centers (*n* = 32, 28.8%), hospitals (*n* = 28, 25.2%), senior welfare centers or dementia daycare centers (*n* = 24, 21.6%), community mental health centers (*n* = 14, 12.6%), and special schools (*n* = 13, 11.7%). The mean age was 55.5 years (SD = 20), and most participants were female (*n* = 88, 79.3%). Participants experienced more depressive symptoms than anxiety symptoms, and 69.4% of participants (*n* = 77) had both symptoms. More than half (*n* = 59, 53.2%) had mental disorders. Participants diagnosed with a mental disorder were classified as follows: 15 (13.5%) for neurodevelopmental disorders, 14 (12.6%) for neurocognitive disorders, 13 (11.7%) for depressive disorders, 12 (10.8%) for schizophrenia, 4 (3.6%) for bipolar spectrum disorders, 1 (0.9 %) for anxiety disorder, and 52 (46.8%) had none of the mental disorders. The average attendance rate for the 30th session was 87% (30–100%, SD = 0.16), and the reasons for absence were personal work, hospital visits, and self-quarantine due to a pandemic.

### 3.1. The Effects of Therapeutic Gardening 

The effect sizes (Cohen’s d), the results of the multilevel analysis, and the mean scores at each time point of the five mental health measures (MHS:D, MHS:A, EDAS, WHOQOL-BREF, and MAAS) are presented in Table 2. The mean scores of all the five mental health measures significantly changed. The mean scores of depression and anxiety decreased, and daily activity, QoL, and mindfulness increased at post-test compared to pre-test. The mean MHS:D score decreased sig-nificantly from 18.2 at baseline to 9.55 at post-test (*B* = −0.998, *p* = 0.0014), with a large effect (*d* = 0.84). The mean MHS:A score decreased significantly from 16.2 at baseline to 8.98 at post-test (*B* = −0.741, *p* = 0.0033) with a medium effect (*d* = 0.72). The mean EDAS score increased significantly from 15.6 at baseline to 18.1 at post-test (*B* = 0.236, *p* = 0.0037), with a medium effect (*d* = 0.61). The mean WHOQOL-BREF score increased significantly from 76.4 at baseline to 87.2 at post-test (*B* = 9.710, *p* = 0.0003), with a medium effect (*d* = 0.64). The mean MAAS score increased significantly from 55.9 at baseline to 62.7 at post-test (*B* = 6.296, *p* = 0.0035), with a small effect (*d* = 0.40). 

### 3.2. Moderation Analysis

#### 3.2.1. Gender

Among the five mental health variables, there was a significant moderating effect of gender on MHS:D (*p* = 0.0427). That is, significant reductions in depression over time were found only for female participants (*B* = −1.157, *p* < 0.0001), but not for male participants (*B* = −0.502, *p* = 0.2097; Figure 1). The results of the moderation analysis of gender for all measures are presented in Appendix A.

#### 3.2.2. Age

Among the five mental health variables, age had a significant moderating effect only on the MHS:D (*p* = 0.0042). While significant reductions in depression over time were found for both adults (*B* = −0.702, *p* = 0.0002) and older adults (*B* = −1.497, *p* < 0.0001), the decrease rates were steeper in older adults than in adults (Figure 1). The results of the moderation age analysis for all the measures are presented in Appendix A.

#### 3.2.3. Mental Disorders

Among the five mental health variables, there was a significant moderating effect of mental disorders only on the WHOQOL-BREF (*p* = 0.0008) and MHS:D (at trend level, *p* = 0.0661). While a significant increase in QoL over time was found for participants without mental disorders (*B* = 15.951, *p* < 0.0001) and participants with mental disorders at the trend level (*B* = 4.474, *p* = 0.059), the increase rates were steeper for participants without mental disorders than for those with mental disorders (Figure 1). Regarding MHS:D, while a significant reduction in depression over time was found for participants without mental disorders (*B* = −1.263, *p* < 0.0001) and participants with mental disorders (*B* = −0.761, *p* = 0.0007), the rate of decrease was steeper for the group without mental disorders than for the group with mental disorders (Figure 1). The results of the moderation analysis for the presence of mental disorders for all measures are presented in Appendix A.

To examine the differences in the effects of therapeutic gardening for each mental disorder, multilevel analyses were performed individually. The results of the multilevel analysis for each mental disorder are presented in Appendix A. The mean MHS:D score significantly decreased over time in patients with depressive and neurocognitive disorders (*p* < 0.05). The mean MHS:A score significantly decreased over time only for depressive disorders (*p* < 0.001). The mean EDAS score significantly increased over time only in patients with schizophrenia (at trend level, *p* = 0.09). The mean WHOQOL-BREF score significantly increased over time only for depressive disorders (at trend level, *p* = 0.06). In the case of MAAS, none of the mental disorders showed significant changes over time.
ijerph-19-08046-t002_Table 2Table 2Effect sizes (Cohen’*d*), the result of multilevel analysis, and mean scores of five mental health measures.
Cohen’s *d*Multilevel Analysis
Mean Scores
Effects EstimateSEdft*p*
T1T2T3T4T5T6T7T8**MHS:D**0.84




Mean18.213.712.412.811.511.310.29.55Intercept
12.5801.5036388.369<0.0001(SD)(8.91)(10.4)(9.72)(11.5)(9.89)(10.8)(9.39)(9.42)Time
−0.9980.230638−4.3400.0014 *N1108198831027299105**MHS:A**0.72




Mean16.213.711.211.811.411.610.48.98Intercept
11.6151.5546397.472<0.0001(SD)(9.42)(10.4)(9.68)(10.5)(10.5)(10.2)(9.83)(9.17)Time
−0.7410.166639−4.4690.0033 *N1118198831027299105**EDAS**0.61




Mean15.615.917.016.716.815.917.018.1Intercept
16.9660.64163826.473<0.0001(SD)(3.81)(3.05)(4.01)(4.10)(4.52)(3.23)(4.37)(4.32)Time
0.2360.0816382.9110.0037 *N1108198831027299105**WHOQOL-BREF**0.64




Mean76.4





87.2Intercept
82.2052.3309135.278<0.0001(SD)(14.2)





(18.1)Time
9.7102.604913.7290.0003 *N98





92**MAAS**0.40




Mean55.9





62.7Intercept
58.9501.6959834.789<0.0001(SD)(15.4)





(15.4)Time
6.2962.106982.9900.0035 *N104





99Note: SE = Standard Error. df = degree of freedom. t = t-value. * *p*-value < 0.01. The interval between time points is 2 weeks (4 sessions). WHOQOL-BREF and MAAS were only assessed at pre- and post-test. T1 = Pre-test, T8 = Post-test; MHS:D—Mental Health Screening Tool for Depressive disorders; MHS:A—Mental Health Screening Tool for Anxiety disorders; EDAS—Engagement in Daily Activity Scale; WHOQOL-BREF—Brief version of WHO Quality of Life; MAAS—Mindful Attention Awareness Scale. 

## 4. Discussion

This study is the first multi-site trial with a large number of participants showing the effects of a therapeutic gardening program for those experiencing psychological distress during the COVID-19 pandemic. As hypothesized, participants demonstrated improvements in all five mental health and well-being variables (depression, anxiety, daily activities, QoL, and mindfulness) over time. These findings are consistent with the positive effects of therapeutic gardening on mental health that have been reported in previous studies [29,39,42,65,66]. In addition, the effect size of depression was large, and anxiety, QoL, and daily activities were medium, which are larger than the overall aggregated effect size of 0.42, as reported in a previous meta-analysis [45]. Considering that the previous meta-analysis analyzed data on physical (e.g., body mass index) and mental health (e.g., depression) variables [45], the current findings would suggest more improvement effects of gardening on mental health and well-being than on physical health. The differential and larger effects of therapeutic gardening on mental health than physical health were also supported by another meta-analysis focusing on nutrition and physical variables with an effect size of 0.18 [67]. Interestingly, in this study, the mean scores for depression, anxiety, and daily activities were significantly changed even after eight sessions. Several previous studies have shown that mental health significantly improved after 8–12 short-term nature-based therapy sessions [33,65,68,69]. Thus, whether short-term therapeutic gardening sessions would lead to significant and equivalent changes in mental health and well-being should be further investigated. We explored the differences in the benefits of therapeutic gardening between gender, age groups, and the presence of mental disorders. The benefits of therapeutic gardening for depression were found only in women and were greater in older adults or those without mental illness. Regarding QoL, participants without mental illness showed greater improvement over time than those with mental illness. As there are few studies on gender differences in the gardening effect, it is difficult to provide an accurate explanation for the insignificant gardening effect on men with depression. One conjecture is that only 21% of the participants in this study were men, and most (70%) had mental disorders. For depression and QoL, therapeutic gardening was found to be more effective for those without mental illness than for those with mental illness, indicating that therapeutic gardening should be provided as an adjunct intervention to already proven effective psychosocial treatments, such as cognitive behavioral therapies, for people experiencing mental disorders. Since the effects of the therapeutic gardening program differed by the type of mental disorder (e.g., greater improvement in depression, anxiety, and QoL among participants with depressive disorders than other groups), it should be further investigated whether the gardening program targeting specific mental illnesses (considering patients’ characteristics and specific needs) would enhance the benefits for people with various mental disorders. Regarding the moderating effects of age, the effect size was greater for older adults. Considering the group activity components of gardening in this study, older adults who are more affected and isolated by the social distancing policy during the COVID-19 pandemic might benefit more than younger ones from the opportunity of social engagement during the gardening program [70,71,72]. For older adults, gardening programs should be more actively implemented given the benefits of improving physical health, emotional stability, and social relationships [73,74,75]. 

This study had some limitations. First, a one-group pre- and post-test design was adopted because of its exploratory nature during the COVID-19 pandemic. Future studies should adopt a controlled design. Second, the durable effects of the therapeutic gardening program are unknown because of the lack of follow-up assessments in this study. Given that some preliminary studies have reported the persistence of the gardening effect at 3–12 months after the end of gardening programs [27,29,68], follow-up assessments should also be included in future studies. 

## 5. Conclusions

Our results suggest that therapeutic gardening programs have been associated with improvements in mental health and well-being for people suffering psychological distress during the long-term COVID-19 pandemic in South Korea, and are particularly effective for women, the elderly, and those without mental illness. Our results regarding high attendance rate (i.e., feasibility) and its potential positive benefits also propose that the highly engaged therapeutic gardening program demonstrates a promising and applicable nature-based intervention for people living in the community. Therefore, in the future, larger and longer-term clinical randomized controlled studies are warranted to arrive at definite conclusions that validate the results, explore other outcomes, identify mechanisms, and investigate the efficacy and effectiveness of therapeutic gardening for participants with various characteristics (gender, age, and presence of mental disorder).

## Figures and Tables

**Figure 1 ijerph-19-08046-f001:**
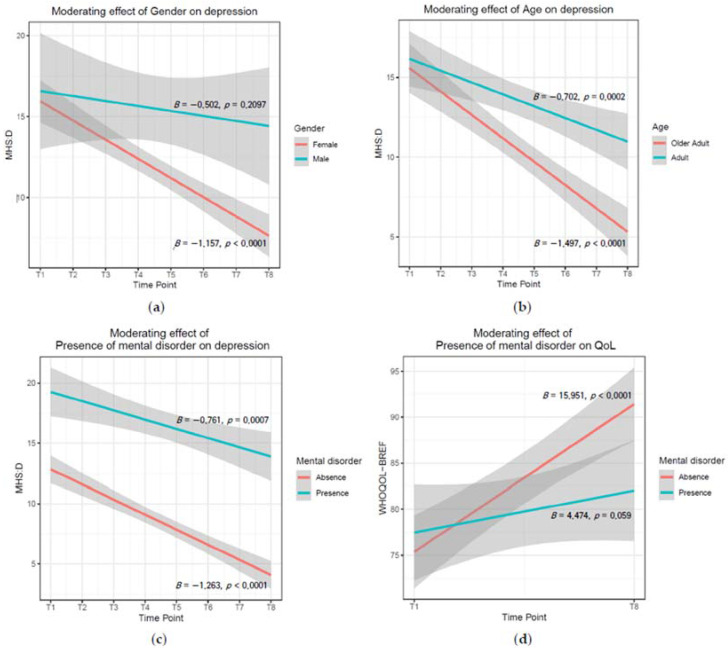
Significant moderating effect of gender, age, and presence of mental disorder for Mental Health Screening Tool for Depressive disorders and Brief version of WHO Quality of Life. (**a**) Moderating effect of gender on depression (*p* = 0.0427); (**b**) moderating effect of age on depression (*p* = 0.0042); (**c**) moderating effect of presence of mental disorder on depression (*p* = 0.0661); (**d**) moderating effect of presence of mental disorder on quality of life (*p* = 0.0008). The interval between time points is 2 weeks (4 sessions). T1 = Pre-test, T8 = Post-test. Shaded areas represent the 95% confidence intervals.

**Table 1 ijerph-19-08046-t001:** Participants Characteristics (*N* = 111).

Variables	*N* (%)
**Gender**	
Male	23 (20.7%)
Female	88 (79.3%)
**Age**	
18–30	17 (15.3%)
31–40	8 (7.2%)
41–50	18 (16.2%)
50–64	25 (22.5%)
≥65	43 (38.7%)
**Mental disorder**	
None	52 (46.8%)
Neurodevelopmental Disorder	15 (13.5%)
Neurocognitive Disorder	14 (12.6%)
Depressive Disorder	13 (11.7%)
Schizophrenia	12 (10.8%)
Bipolar Spectrum Disorder	4 (3.6%)
Anxiety Disorder	1 (0.9%)
**Education**	
No Education	1 (0.9%)
Primary school	8 (7.2%)
Secondary school	18 (16.2%)
University bachelor degree	53 (47.7%)
Higher Education (>15 years)	8 (7.2%)
Unknown	23 (20.7%)

Note: *N* = number of participants.

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
