# Peer review of "The Multi-Sites Trial on the Effects of Therapeutic Gardening on Mental Health and Well-Being"

_ijerph, 2022, doi:10.3390/ijerph19138046_

Round 1

Reviewer 1 Report

It is a very interesting research, although somewhat simple to read. It corresponds to a current research topic and may provide a different point of view to the literature published so far on the effects of therapeutic gardening. Therefore, in my opinion this research should be published with a few small remarks: The introduction is short and does not provide an entry point to the topic to be developed, moreover the objectives are not clearly defined. Finally, the conclusion does not describe the main contributions of the analysed topic. 

Reviewer 2 Report

This manuscript is well written and organized, and can contribute to the literature. Particularly, the authors should be applauded for the longitudinal data collection. Despite the positive aspects of the paper, I provided my comments below, which I hope helps the authors improve the manuscript.

1. It would be helpful to clarify what the percentages shown in the line 53 mean.

2. I wonder whether participants’ mental health and well-being came from COVID-19. It seems like the authors didn’t measure mental health and well-being caused by COVID-19, but during COVID-19. If so, the authors should add statements that justify why it’s theoretically and practically important to examine people’s mental health and well-being during COVID-19, even though those symptoms didn’t stem from COVID-19. I’d like to see stronger justifications on the significance of examining mental health and well-being during COVID-19. I’m asking this because I’m not sure whether COVID-19 matters in the research context, because I don’t see how COVID-19 influences gardening sessions. Please clarify this in Introduction and Discussion.

3. Please provide more information about the participant recruitment process in terms of how the authors could get access to the participants, any compensation etc.

4. Line 208, I’m not sure why it’s in bold.

Round 2

Reviewer 2 Report

Thank you for the revision.